# PathSeqSAM: Sequential Modeling for Pathology Image Segmentation with SAM2

**Mingyang Zhu**[1]                                                   STJU18070285728@SJTU.EDU.CN
[1] *Shanghai Jiao Tong University, Shanghai, China*
**Yinting Liu**[2]                                                             YILIU@UNMC.EDU
[2] *University of Nebraska Medical Center, Omaha, USA*
**Mingyu Li**[3]                                                 MINGYU.LI@VANDERBILT.EDU
**Jiacheng Wang**[3]                                       JIACHENG.WANG.1@VANDERBILT.EDU
[3] *Vanderbilt University, Nashville, USA*

**Editors:** Under Review for MIDL 2025

## Abstract

Current methods for pathology image segmentation typically treat 2D slices independently, ignoring valuable cross-slice information. We present PathSeqSAM, a novel approach that treats 2D pathology slices as sequential video frames using SAM2's memory mechanisms. Our method introduces a distance-aware attention mechanism that accounts for variable physical distances between slices and employs LoRA for domain adaptation. Evaluated on the KPI Challenge 2024 dataset for glomeruli segmentation, PathSeqSAM demonstrates improved segmentation quality, particularly in challenging cases that benefit from cross-slice context. We have publicly released our code at https://github.com/JackyyyWang/PathSeqSAM.
**Keywords:** Pathology image, SAM2, cross-slice attention, glomeruli segmentation

## 1. Introduction

Accurate segmentation of histopathological structures is fundamental for quantitative analysis of kidney pathology images, particularly in chronic kidney disease diagnosis (Deng et al., 2024, 2023). Traditional approaches often process each 2D pathology slice independently, overlooking potentially valuable contextual information from adjacent slices of the same specimen. This limitation becomes especially apparent in challenging cases with staining inconsistencies or complex pathological changes (Ginley et al., 2019; Altini et al., 2020).

Recent advances in foundation models, particularly SAM2 (Ravi et al., 2024), have shown promise in handling sequential data, but directly applying these methods to pathology remains challenging due to domain shift (Wu et al., 2023; Li et al., 2024; Ma et al., 2024). We propose PathSeqSAM, which interprets multiple 2D slices from the same subject as sequential video frames, enabling cross-slice contextual learning through:

- A sequential modeling strategy that treats pathology slices as video frames.

- A distance-aware attention mechanism to accommodate variable physical distances.

- Domain adaptation using Low-Rank Adaptation (LoRA) (Hu et al., 2021) for pathology-specific features.

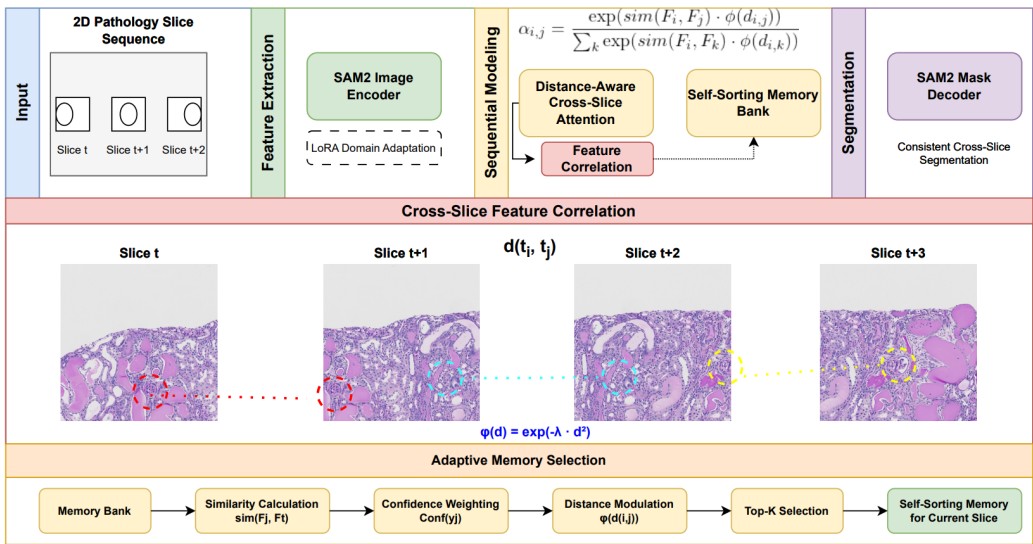

Figure 1: Proposed SAM2 segmentation pipeline for pathology images. Features extracted via the LoRA-adapted SAM2 encoder are refined by distance-aware cross-slice attention and a memory bank, enabling coherent segmentation across slices.

## 2. Methods

**Problem Formulation:** Given a set of 2D pathology slices $\{S_1, S_2, ..., S_n\}$ from the same subject, we formulate a sequential segmentation task, where each slice functions as a frame in a video-like sequence. This viewpoint captures cross-slice relationships despite differences between typical video frames and pathology slices. As shown in Figure 1, our approach leverages the correspondence between pathology slices, where structures like glomeruli can be tracked across sequential slices, similar to objects moving through video frames.

**Distance-Aware Cross-Slice Attention:** SAM2 extends the original SAM (Kirillov et al., 2023) by incorporating a memory attention mechanism to handle sequential data. However, pathology slices often have variable physical distances, unlike uniformly spaced video frames. To address this, we introduce a distance-aware attention mechanism:

$$\alpha_{i,j} = \frac{\exp\big(sim(F_i, F_j) \cdot \phi(d_{i,j})\big)}{\sum_k \exp\big(sim(F_i, F_k) \cdot \phi(d_{i,k})\big)}, \tag{1}$$

where $F_i$ and $F_j$ are feature embeddings for slices $i$ and $j$, $sim(\cdot, \cdot)$ is a cosine similarity function, $d_{i,j}$ is the estimated physical distance, and $\phi(d) = \exp(-\lambda \cdot d^2)$ is a distance modulation function. The parameter $\lambda$ is initialized to 0.1 and learned during training to adaptively weight the influence of physical distance on attention.

**Adaptive Memory for Histopathology Context:** We adopt an adaptive slice selection strategy instead of maintaining a fixed-size memory of recent frames, choosing the most informative slices based on feature similarity and confidence:

$$M_t = \{F_j \mid j \in \text{top-K}\big(sim(F_j, F_t) \cdot Conf(y_j)\big), j < t\}. \tag{2}$$

Table 1: Patch-level Diseased Glomeruli Segmentation Performance (Dice Score %)

| Method | Mean±SD |
| --- | --- |
| nnUNet | 88.79±5.38 |
| Swin-Unet | 89.65±6.41 |
| SAM2 | 92.48±6.13 |
| PathSeqSAM (Ours) | **94.71±5.89** |

Here, $Conf(y_j)$ represents the segmentation confidence derived from SAM2's cross-attention mechanism. This approach prioritizes slices with higher feature similarity and reliable segmentation confidence, regardless of strict sequential ordering. Such flexibility is critical for pathology slides, where the relationship between adjacent slices can be complex.

**Domain Adaptation:** We utilize Low-Rank Adaptation (LoRA) (Hu et al., 2021) to adapt SAM2's image encoder for the pathology domain. LoRA injects trainable low-rank matrices into attention layers, preserving most pre-trained weights. This method effectively handles domain adaptation with minimal computational overhead (Cheng et al., 2023).

## 3. Experimental and Results

We implemented PathSeqSAM on the SAM2 codebase (Ravi et al., 2024), applying LoRA with rank 8 to the image encoder. The model was trained on the KPI Challenge 2024 dataset (Deng et al., 2024) using a combined loss function:

$$\mathcal{L} = \mathcal{L}_{dice} + 0.5\,\mathcal{L}_{BCE} + 0.2\,\mathcal{L}_{consistency}, \tag{3}$$

where $\mathcal{L}_{consistency}$ encourages consistent segmentation across similar slices by penalizing discrepancies in predictions between slices with high feature similarity (Ji et al., 2021). We set $K = 5$ to balance contextual information and computational efficiency. Physical distances were obtained from metadata or estimated via feature similarity.

Table 1 compares PathSeqSAM with state-of-the-art methods on patch-level glomeruli segmentation from the KPI Challenge 2024. PathSeqSAM achieved a mean Dice score of 94.71±5.89, outperforming nnUNet, Swin-Unet, and SAM2. The 2.23% improvement over SAM2 demonstrates the effectiveness of sequential modeling and distance-aware attention in pathology segmentation.

## 4. Discussion and Conclusion

PathSeqSAM introduces a sequential modeling paradigm for pathology image segmentation by leveraging SAM2's memory attention across multiple slices. The distance-aware attention mechanism and LoRA-based domain adaptation address the unique challenges of histopathological data, such as variable inter-slice spacing and staining inconsistencies. By prioritizing the most informative slices, our adaptive memory approach further enhances segmentation consistency and accuracy.

## Acknowledgments

We thank the organizers of the KPI Challenge 2024 for providing the dataset.

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
