# OpenReview forum: "PathSeqSAM: Sequential Modeling for Pathology Image Segmentation with SAM2"
_MIDL.io/2025/Short_Papers — MIDL 2025 - Short Papers_

### Official Review · Reviewer_Waih · 2025-04-29

**Rating:** 1
**Confidence:** 5

**Summary:**

This paper proposes to add an ability to SAM2 to treat additional images as context, and help to annotate the pathology slices (potentially) better.

**Strengths:**

Interesting idea (though exploited in classification).

**Weaknesses:**

* No details on the data
* No details on the IRB approvals
* Poor statistical evaluation quality
* No baselines that would add a temporal dimension to e.g. UNet

---

### Decision · Program_Chairs · 2025-05-01

**Decision:**

Accept

**Comment:**

The PC discussed the paper during the panel meeting and decided to accept. The PC had a closer look at the raised concerns and decided that given the scope of the short paper track, the short paper provides a sufficient contribution, adequate detail and experimental evaluation.